# Carbon Nanotori Structures for Thermal Transport Applications on Lubricants

**DOI:** 10.3390/nano11051158

**Published:** 2021-04-29

**Authors:** Jaime Taha-Tijerina, Karla Aviña, Juan Manuel Martínez, Patsy Yessenia Arquieta-Guillén, Marlon González-Escobedo

**Affiliations:** 1Departamento de Ingeniería, Universidad de Monterrey, Av. Ignacio Morones Prieto 4500 Pte., San Pedro Garza García 66238, NL, Mexico; karla.avina@udem.edu (K.A.); juan.martineza@udem.edu (J.M.M.); 2Department of Manufacturing and Industrial Engineering, University of Texas Rio Grande Valley, Brownsville, TX 78520, USA; 3Facultad de Ciencias Físico-Matemáticas, Universidad Autónoma de Nuevo León, San Nicolás de los Garza 66455, NL, Mexico; patsy.arquietagl@uanl.edu.mx; 4Facultad de Ciencias Biológicas, Universidad Autónoma de Nuevo León, San Nicolás de los Garza 66455, NL, Mexico; marlon.gonzalezes@uanl.edu.mx

**Keywords:** thermal conductivity, nanotori, lubricants, water-based, oil-based

## Abstract

Carbon nanostructures have been recently applied to improve industrial manufacturing processes and other materials; such is the case for lubricants used in the metal-mechanic industry. Nanotori are toroidal carbon nanostructures, obtained from chemical treatment of multi-wall carbon nanotubes (MWCNTs). This material has been shown to have superb anti-wear and friction reduction performance, having the ability to homogeneously disperse within water in concentrations between 1–2 wt.%. Obtained results of these novel nanostructures under water mixtures and novel additives were a set point to our studies in different industrial lubricants. In the present work, nanotori structures have been applied in various filler fractions as reinforcement to evaluate the behavior in thermal transport of water-based (WB) and oil-based (OB) lubricants. Temperature-dependent experiments to evaluate the thermal conductivity were performed using a thermal water bath ranging from room temperature up to 323 K. The obtained results showed a highly effective and favorable improvement in the heat transport of both lubricants; oil-based results were better than water-based results, with nanotori structures increasing the lubricants’ thermal transport properties by 70%, compared to pure lubricant.

## 1. Introduction

In industrial manufacturing systems a remarkable search to obtain the suitable material application and performance, optimizing devices, machines and designs, maintain or reducing materials consumption and pollution mitigation. Thermal transport plays a vital role in industrial manufacturing systems and due to global market competition and cost reduction endeavors, design miniaturization has required improvements in the thermal performance (heat dissipation efficiency) of lubricants and fluids [1]. 

Lubricants and fluids are required in numerous engineering applications and fields such as energy, transportation, defense, aerospace, microelectronics, power transmission, and nuclear systems cooling, among others [2,3,4], for thermal transport and reduction of wear and friction in metal-mechanic manufacturing operations, such as in forming-punching, machining, drilling, among others [5]. These materials provide a proper working component interface, removing chips and debris from machined surfaces, reducing the risks associated with machinery failure or tooling damage, improving the quality of working components.

Early investigations used millimeter or micrometer-sized particle suspensions in order to improve material performance. Nevertheless, initially, this led to a range of issues such as a rapid sedimentation tendency of these solid structures within the fluidic media; forming sludge sediments; limiting the thermal transport capacity; and increasing the thermal resistance of the conventional fluids and lubricants. 

With the aid of nanotechnology, diverse heat transfer fluidic media have been investigated in order to improve thermal transport behavior, characteristics and performance [6,7,8,9,10,11,12,13]. Nanofluids, and some nanolubricants, are highly effective thermal transport media as a result of their anomalous high thermal characteristics. Solid nanostructures possess high thermal conductivity, measuring several orders of magnitude higher than conventional heat transfer lubricants, fluids, and mixtures of both. Integration of reinforcing nanostructures within conventional materials results in higher thermal conductive mixtures that could enhance performance and efficiency in diverse systems [14]. Among the benefits of nanofluids and nanolubricants, one of the attractive characteristics is that these materials can be optimally engineered to achieve particular objectives, such as anti-wear properties, thermal energy storage capacity, high thermal conductivity, better temperature stabilization and lower pressure drop [15]. Nevertheless, even though nanofluids and nanolubricants are promising for many practical applications, care must be taken to maintain the integrity of the systems and eliminate undesirable properties. Potential drawbacks of adding nanoreinforcements to conventional materials pertain to viscosity, nanostructures’ agglomeration, and sedimentation, among others [15,16,17].

A carbon nanotube (CNT) is a carbon-based material in tubular shape, a few nanometers in diameter. Carbon-based nanostructures can be produced in a variety of sizes and lengths. They resemble black powder on a macro scale. Among a wide range of nanostructures, varying from low to very high thermal conductivity, carbon-based materials have been shown to provide the highest thermal conductivity improvements compared to other particle types. Experiments have been conducted on carbon-based nanostructures such as graphene, carbon fibers, and CNTs, in order to increase thermal transport performance of conventional materials [18,19,20,21,22,23,24,25]. In diverse studies, nanomaterials developed with allotropes of carbon (graphene, diamond, CNTs, carbon fibers, etc.) and oxide-based materials generally contain more than 1.0 wt.% of these reinforcements’ concentration [9,26,27,28,29]. Nanofiller concentrations up to up to 10–12 wt.% have been incorporated into conventional media to enhance the system’s thermal characteristics (typically by at least 12% in thermal conductivity). However, such concentrations of nanostructures showed various negative effects, such as an increase in the suspension’s viscosity, as well as cost. The higher concentrations of nanofillers adversely affect the material fluidity, threaten the nanofluid´s stability, and thermal management goals. 

Choi et al. [30] reported superb thermal conductivity improvements for multi-wall CNTs (MWCNTs) reinforcing synthetic poly (α-olefin) oil (PAO). They achieved enhancement of 160% at 1.0 vol.% of MWCNTs. The effects of solid nanostructures at various volume fractions and temperatures on thermal transport performance of hybrid CNTs–Al_2_O_3_/water nanofluids was investigated by Esfe et al. [31], they found that thermal conductivity improvement was due to the high number of solid reinforcements and rise in evaluating temperatures. Farbod et al. [32] prepared water-based nanofluids at various concentrations and lengths of functionalized MWCNTs. An improvement in thermal conductivity was observed with increasing the temperature and with decreasing the length of the MWCNTs, suggesting a percolation channel formation and interlayer interactions between the carbon nanostructures and the media. A study performed by Ilyas et al. [33] on commercial brand oil with MWCNTs reinforcements, showed a significant thermal conductivity enhancement at 1.0 wt.% of 22 and 33% at 35 and 333 K, respectively. Naddaf et al. [23] observed significant enhancements in thermal conductivity (>80%, when compared to pure diesel oil) with the increase in temperature, up to 373 K, for graphene nanoplatelets- and MWCNT/diesel oil nanofluids. Similarly, Shanbedi et al. [34] observed that increasing the filler fraction and temperature of their studied water-based nanofluids, an improvement in the effective thermal conductivity performance was achieved. Graphene nanoplatelets were dispersed within water and ethylene glycol (EG) by Selvam et al. [35]. It was observed that as the filler fraction of graphene nanoplatelets was increased, the thermal conductivity ratio was enhanced. Particularly, an improvement of 20% and 15%, for EG and water media was observed at 0.5 vol.%, respectively. On other carbon structures, Branson et al. investigated nanodiamonds (NDs) dispersed within EG and mineral oil (MO) [26]. In their studies, incorporation of 0.88 vol.% of NDs increased the thermal conductivity by ~12%. Moreover, 6% and 11% increases were achieved at 1.0 and 1.9 vol.%, respectively. Similarly, Yu et al. [36] showed that incorporation of small concentrations (~1.0 vol%) of NDs to EG produced thermal conductivity enhancements of around 17%. In previous work by Taha-Tijerina et al. [8], even less filler fraction of NDs (0.1 wt.%) was homogeneously dispersed within MO-based nanofluids with significant improvements of ~70%.

Among these carbon-based materials, less-common structures have been recently studied for mechanical applications and other functions. These novel material configurations are called nanotori or nanorings, comprising of circular CNTs formed by connecting their two ends. The toroidal-like structures are closely related to the well-known sphere-like nanostructures. As mentioned by Kharissova et al. [37], these torus-like materials have been studied more theoretically than experimentally. However, scientific studies have shown their superb properties and promising applications. Recently, carbon nanotori structures have been evaluated as reinforcements for conventional lubricants and fluids in the metal-mechanic field, allowing a reduction in wear and friction [38]. 

Due to their exceptional characteristics and similarity to CNTs, carbon nanotori structures are proposed as reinforcing material for metal-forming lubricants. In this research, water-based (WB) and oil-based (OB) nanolubricants consisting of varying carbon nanotori filler fractions (by weight) aim to determine their thermal transport performance (thermal conductivity) over a range of temperatures (up to 323 K).

## 2. Materials and Methods

### 2.1. Preparation of Nanolubricants

In our research, two commonly applied lubricants (FUCHS Industry), water-based and oil-based, for metal-forming applications; punching, stamping, and drawing, among others, are used as conventional materials (see Table 1). Their general characteristics provide a light but effective film to dissipate heat and reduce wear in tools and machinery. These materials were reinforced with carbon nanotori at various filler fractions: 0.01 wt.%, 0.05 wt.% and 0.10 wt.%. Multi-walled carbon nanotubes (MWCNTs) (30–50 nm in diameter) functionalized with carboxylic acid (COOH) groups were dissolved in an acid solution of nitric acid (HNO_3_), sulfuric acid (H_2_SO_4_), hydrochloric acid (HCl), potassium permanganate (KMnO_4_), hydrogen peroxide (H_2_O_2_), and deionized water (DiW) [39]. Afterwards, magnetic stirring was performed to the solution in a glass container in an ice bath for 48 h. Then, the solution was moved to a glass container where DiW was poured at 20% concentration and maintained for another 24 h (Figure 1a). HCl and H_2_O_2_ were used to clean the solution which is centrifuged afterwards to obtain the supernatant which contains the carbon nanotori. Figure 1b shows a TEM (Hitachi H-9500, Tokyo, Japan, electron microscope operated at 300 keV) image of carbon nanotorus suspended in DiW; these nanostructures have a torus diameter of ∼130 nm, and tube diameter of ∼40 nm. Dynamic light scattering (DLS) (Malvern Zetasizer Nano ZS, Almelo, The Netherlands) was used to determine that the average outer diameter of these nanostructures was ~177 nm with a polydispersity index of 0.248 and zeta potential of −26.5 mV [38]. 

Raman spectroscopy for nanotori structures (Raman: Micro-RAMAN DXR Thermoscientific, Sussex, UK at 532 nm) is shown in Figure 2. Synthesis was carried out using MWCNTs, whose characteristic Raman spectra contain the G band at 1590 cm^−1^, (the same spectra of almost all carbon nano forms); the D band around at 1320 cm^−1^ indicates a structural defect of graphene; and the band at 2700 cm^−1^ G’ (also known as 2D) is used to get information about the electronic and geometrical structure. These spectra allow us to confirm that nanotori structures are conformed from MWCNTs in toroidal shape. Additionally, the increment at 1600 cm^−1^ allows us to know that changes exist in the geometric structure. This increment in Raman absorption reflects a change in crystallinity of the material.

Once carbon nanotori structures were obtained, they were dispersed within WB and OB lubricants. Then, 40 mL glass vials were prepared at various filler fractions: 0.01 wt.%, 0.05 wt.% and 0.10 wt.% of nanotori. To homogeneously disperse the nanostructures, an ultrasonic dismembrator was used (Branson ultrasonic homogenizer model 5510-40 kHz) for a prolonged time (5–6 h). To prevent any agglomeration and fast sedimentation of nanostructures, the sonicator bath water temperature was maintained constant at room temperature (297 K), and the bath water was changed every hour. Vials were kept on a shelf for at least 4 weeks without significant visible settling or sedimentation. UV–Vis spectra of carbon nanotori WB lubricant at 0.10 wt.% is shown in Figure 3. Overlap of as prepared (0 h) sample (black) and sample after 10 weeks are presented with no significant structural changes, confirming what we observed regarding the stability of the suspensions after a long shelf-sitting time.

### 2.2. Thermal Conductivity Measurements

There are diverse methodologies and techniques to determine and evaluate the thermal conductivity of fluidic media: transient hot-wire technique [14,40,41], cylindrical cell method [42], steady-state method [43,44], temperature oscillation method [45], and 3-ω method [46,47]. In our research, thermal evaluation was performed by thermal conductivity measurements of conventional WB and OB nanolubricants at various carbon nanotori filler fractions according to the transient hot-wire (THW) methodology, with a KD2 Pro instrument (Decagon, Inc., Pullman, WA, USA). This technique obtains the thermal conductivity of a fluidic media which is determined based on measuring the time response and temperature of a metal probe subjected to an immediate electrical pulse, which functions both as a heater and temperature sensor. The KD2 Pro works with stainless-steel probe (KS-1; 1.3 mm in diameter by 60 mm long) which is fully immersed in the vial sample to obtain the effective thermal conductivity of nanolubricants. The probe temperature is slowly increased by providing a constant current through resistive heating; the heat is dissipated from the sensor to the surrounding media through conduction, hence, increasing its temperature. This temperature rise depends on the media’s thermal conductivity. Finally, the thermal conductivity value is obtained from the heating power and the slope of the temperature change through the device algorithm using a logarithmic time scale. The transient hot-wire method has the advantages of fast measurements and low cost, which increases the ease of implementation. The system was calibrated/verified before the measurements were taken, using a standard fluid (glycerol) and corroborated data for DiW and EG. The thermal conductivity values are verified up to 3 decimal points. Enhancements in thermal conductivity were obtained, considering the ratio of effective thermal conductivity of the nanofluids (*k_eff_*) and pure lubricants, WB and OB lubricants, respectively, (*k*_0_) and ((*k_eff_*/*k*_0_) − 1)%.

Temperature-dependent measurements were performed using a water bath ranging at various temperatures from room temperature (297 K) up to 323 K (See Figure 4). Each sample was thermally equilibrated before each measurement for 10–15 min. Average thermal conductivity values were taken with standard deviation from at least 6 measurements from each set of nanolubricants.

## 3. Results

Improvements in thermal conductivity for carbon nanotori within OB and WB lubricant systems have been obtained. Figure 5 shows the temperature-dependent thermal conductivity performance of these systems at various carbon nanotori filler fractions; error bars depict the standard deviations.

Pure WB lubricant did not show considerable temperature dependency (less than 2% at 323 K). All WB nanolubricants showed gradually thermal conductivity increments as carbon nanotori concentrations and temperature were also increased, indicating the thermal behavior of carbon-based nanostructures. Moreover, the incorporation of carbon nanotori produced important thermal conductivity enhancements as evaluating temperature was increased. For instance, at 323 K, WB nanofluids achieved enhancements of 24, 33 and a maximum of 46% at 0.01 wt.%, 0.05 wt.% and 0.10 wt.%, respectively, when compared to pure WB lubricant.

Similar to WB lubricants, OB lubricants did not show temperature dependency (less than 2% at 323 K). Thermal conductivity evaluations were performed as filler fractions and temperatures were increased, to determine their temperature-dependency behavior. The incorporation of carbon nanotori reflected significant improvements as temperature was increased. At 313 K for instance, OB nanofluids achieved enhancements of 16%, 24% and 36%, at 0.01 wt.%, 0.05 wt.% and 0.10 wt.%, respectively. Superb behavior observed was achieved at 323 K, with maximum enhancement of 70% at 0.10 wt.%.

Incorporation of carbon nanotori, even at very low filler fractions (up to 0.10 wt.%), significantly enhances the thermal conductivity performance of nanolubricants, which is mainly attributed to the inherent high heat transfer capacity of carbon-based nanostructures.

Due to the low applied nanostructures concentrations, the resulting improvements in thermal conductivity could be attributed to diverse factors, such as molecular interactions between the lubricants and carbon nanostructures [18,22,25,48], and percolation mechanism [49,50,51]. As the nanotori filler fraction is increased within the lubricants, the nanostructures’ distance is decreased, thus increasing the contact probability among them; therefore, thermal transport channels are formed, increasing the thermal conductivity behavior due to the percolation mechanism [52]. Another important factor is the Brownian motion contribution of the carbon nanostructures [51,52,53,54]. For instance, an increase in the thermal conductivity of the nanolubricants can be induced by the heat transport between colliding nanostructures, particularly at higher temperatures, corresponding to more intense Brownian motion [55,56,57]. Furthermore, liquid layering at the lubricant/nanotori interface could also contribute to the increased behavior of thermal conductivity [58,59,60]. Effective thermal conductivities (*k_eff_*) of nanolubricants increase with temperature (room temperature up to 323 K), indicating the role of Brownian motion on measured thermal conductivities, in accordance with Maxwell’s predictions [61,62]. 

Theoretical approaches and correlations used to describe and explain improvements in thermal transport performance for nanomaterials is so complex that the use of a single model is not sufficient to predict the wide range of experimental data [63]. Theoretical models have been used to predict thermal conductivity of nanofluids assuming diverse variables, such as the reinforced nanostructures properties and characteristics being well-dispersed within conventional media. 

The filler fractions and conductivities of the lubricants will determine the lower and upper boundary values of nanolubricants’ effective thermal conductivity (*k_eff_*). A theoretical model of the *k_eff_* of the nanolubricants is performed using a classical effective medium theory known as Hashin-Shtrikman (H-S) theory [64]. In this case the ratio of *k_nt_*/*k_L_* is >1, where *k_nt_* is the thermal conductivity of the carbon nanotori and *k_L_* is the thermal conductivity of the lubricants. The lower boundary value (since the nanofillers fraction, *φ*, is very low, 0.10 wt.%) for effective thermal conductivity of nanolubricants, *k_eff_*_,_ is calculated using the following equation:
(1)$$\frac{k_{eff}}{k_{L}}=1+\frac{3\varphi \left(\frac{k_{nt}}{k_{L}}-1\right)}{\frac{k_{nt}}{k_{L}}+2-\varphi \left(\frac{k_{nt}}{k_{L}}-1\right)}$$

The calculated value 0.5486 W/m K matches well with the experimental value obtained at room temperature for 0.10 wt.% nanolubricant (0.547 W/m K). The model from Hashin-Shtrikman (H-S) requires input of the reinforcing nanostructures’ thermal conductivity. Carbon nanotori allotropy has not been fully described in literature for its thermal performance. However, carbon nanostructures (graphite, graphene, SWCNTs, MWCNTs) have been reported at various thermal conductivities, varying from 2000 up to 6000 W/mK [65,66,67,68,69]. For our evaluations, we selected a lower value considering the structural configuration (ring/tori) of the nanostructure in our research. In this case, carbon nanotori structures were considered to have a thermal conductivity of 2200 W/mK, which was applied as *k_nt_* value.

From a theoretical point of view, with the increase of the nanofluid’s bulk temperature, molecules and nanostructures are more active and able to transfer more energy from one location to another per unit time. The temperature-dependent variations in thermal conductivity indicate that it is not just the percolation mechanism that increases the thermal conductivity, but also Brownian motion contributes to the thermal transport behavior of carbon nanotori-based nanolubricants as well.

## 4. Conclusions

Reinforcing conventional materials with solid carbon nanotori structures, promotes highly effective heat transfer behavior, which is mainly attributed to its anomalous high thermal conductivity at very low concentrations up to 0.10 wt.%. Two conventional metal-forming lubricants were analyzed, water-based and oil-based. For oil-based lubricants, thermal conductivity improvements were observed as carbon nanotori and evaluating temperatures were increased, reaching a maximum of 46% at 323 K with merely 0.10 wt.%. The greatest impact was shown with oil-based lubricants, probably due to the oleophilic compatibility of carbon nanostructures. Here, the enhancement was observed to achieve 36% at 313 K. The maximum improvement was shown at 323 K up to 70% for 0.10 wt.% filler fraction of carbon nanotori. Carbon nanostructures are shown to have good compatibility with conventional lubricants and may significantly improve thermal conductivity when used as reinforcements. Considering the suitability of the reported carbon-reinforced nanolubricants for thermal industrial manufacturing processes, the need of predictive methodologies for thermal transport is an area of opportunity for further development and study.

## Figures and Tables

**Figure 1 nanomaterials-11-01158-f001:**
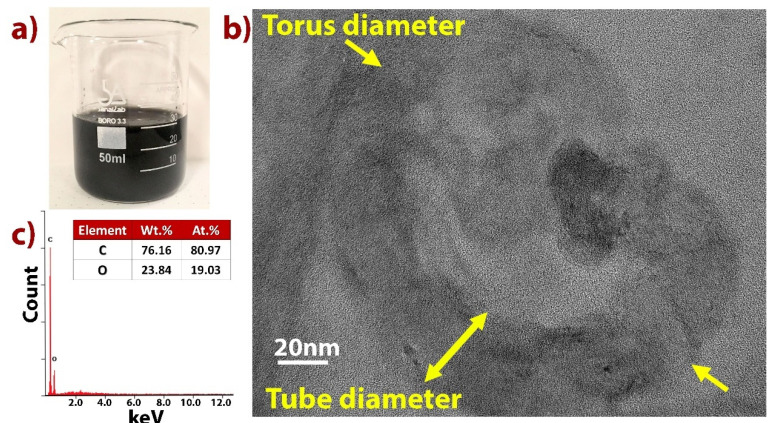
(**a**) Carbon nanotori structures suspended within DiW; (**b**) TEM of analyzed carbon nanotori; and (**c**) EDAX analysis of carbon nanotori.

**Figure 2 nanomaterials-11-01158-f002:**
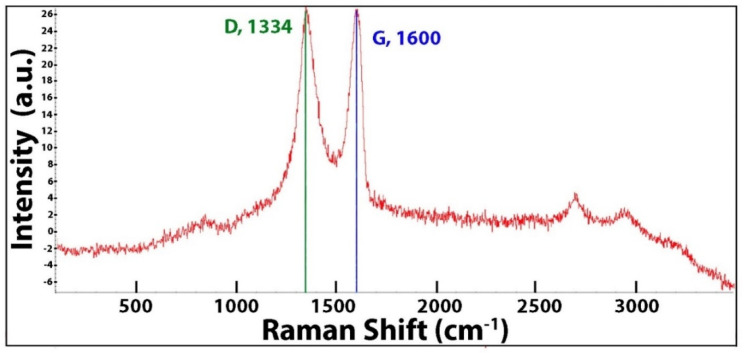
Raman spectrum for carbon nanotori structures.

**Figure 3 nanomaterials-11-01158-f003:**
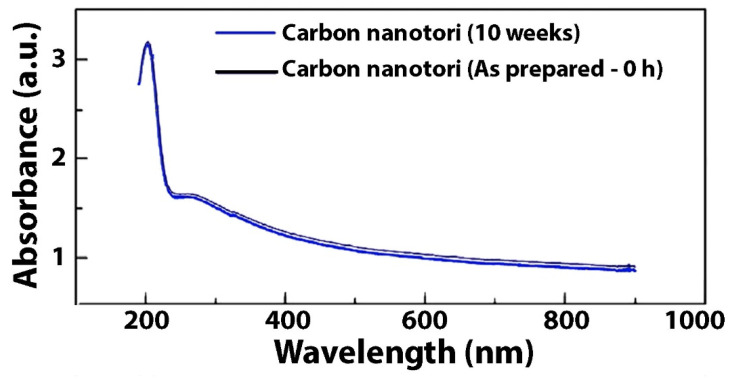
UV–Vis spectra of carbon nanotori WB lubricant at 0.10 wt.%.

**Figure 4 nanomaterials-11-01158-f004:**
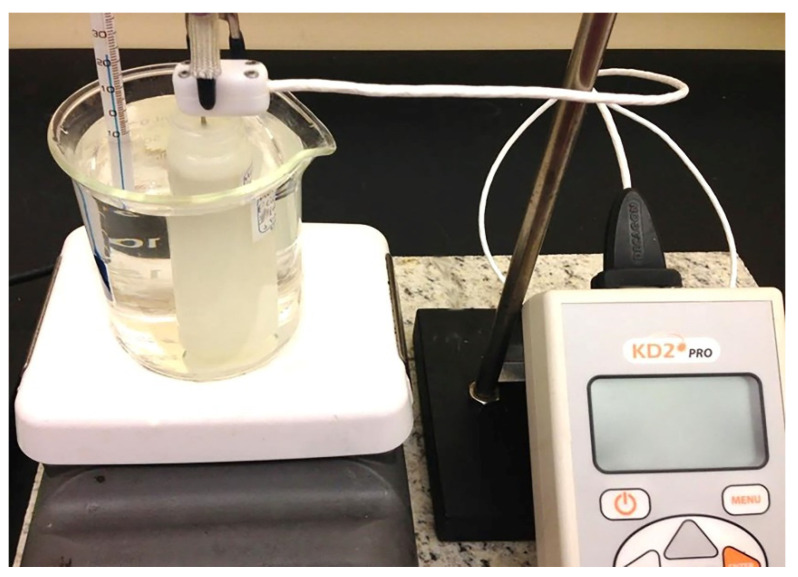
Illustration of configuration used for thermal evaluation for nanolubricants.

**Figure 5 nanomaterials-11-01158-f005:**
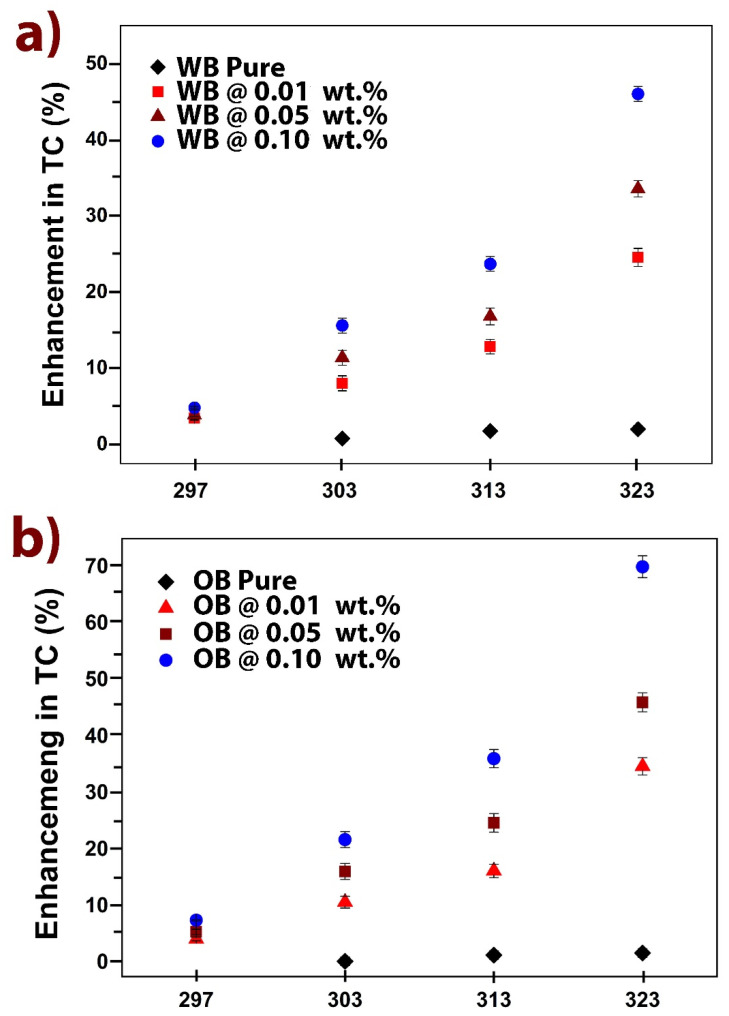
Thermal conductivity enhancements under temperature-dependence evaluation. (**a**) water-based lubricants; (**b**) oil-based lubricants (percentage of filler amount is mentioned).

**Table 1 nanomaterials-11-01158-t001:** Material characteristics.

Materials	Properties		
Base lubricant	Density (288 K)	Kinetic Viscosity (mm^2^/s)	Waer:Oil Ratio
Water-based	1.031 g/cm^3^	2.02 @ 298 K 0.86 @ 343 K	6:1
Oil-based	0.994 g/cm^3^	63.3 @ 298 K 40.7 @ 343 K	4:1
Nanoparticles	Size/Geometry	Polydispersity Index	Zeta Potential
Carbon nanotori	Tube diameter ∼40 nm Torus diameter ∼130 nm	0.248	−26.5 mV
	OD~177 nm		

## Data Availability

The data obtained and utilized in this research has been shown in the figures and graphics, that is all we have to share.

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
