# Peer review of "Carbon Nanotori Structures for Thermal Transport Applications on Lubricants"

_nanomaterials, 2021, doi:10.3390/nano11051158_

Round 1

Reviewer 1 Report

The manuscript only prepared the nanolubricants and obtained the temperature-dependent thermal conductivity performance of these systems at various carbon nanotori filler fractions, which are not enough. Without the study of this manuscript, we still know that the thermal conductivity increases the temperature and concentration which is common objective law.  The manuscript has no any research significance, also has no innovation. Hence, the manuscript cannot be published in this Journal.

Author Response

Dear reviewer,

Thank you very much for all the comments, please see attachments.

Reviewer 2 Report

Dear authors,

Thank you for your submission.

The thermal conductivity properties of water-based and oil-based lubricants that use carbon nanotori structures as additives was investigated in this work. Addition of the nanostructures improved the thermal transport of both lubricants significantly, owing to their great heat transfer capacity. The improvement was found to be affected by the solvent temperatures, i.e. presenting a temperature-dependent behaviour. The following is the list of comments for your reference.

1. Page 2 Lines 87-90. There should be a space between the number and an abbreviated unit, and the unit of the Zeta potential is missing. Same comment goes to Table 1 and the full manuscript.

2. Page 2 Line 95. The carbon nanostructures possess a good dispersion stability, with a Zeta potential of -26.5 mV, in Di-water. Do they have similar or better dispersion in the both commercial water-based and oil-based lubricants?

3. Section 2.2. Authors might need to provide a photo view and/or an illustration of the configuration that use for thermal evaluation for both WB and OB nanolubricants.

Author Response

Dear Reviewer,

Thank you very much for all the comments, please see the attachment.

Reviewer 3 Report

This article experimentally evaluates the use of carbon nanotori structures to enhance the thermal and lubricant performance of water and oil based fluids. Even if somewhat reduced, the investigation is quite novel and results could be interesting for different metal-forming applications. In my opinion, the article may be considered for acceptance in Nanomaterials after authors consider the following comments.

  • Do not use “º” symbol with absolute temperatures (Kelvin). Do not mix “ºC” and “K” temperatures throughout the manuscript.
  • Even if nanotori structures (and their use as thermal conductivity/lubricant nano-enhancers) are still a quite novel topic, the literature revision presented in the introduction should be improved.
  • Indicate the provenance and purity of reagents used in the investigation.
  • Include complete information (brand, model) of all experimental devices.
  • Microscopy analysis is somewhat reduced. If possible, could authors also include another image at a lower magnification?
  • Have authors considered to repeat Zeta potential measurements throughout time?
  • Support the following sentence with an appropriate reference “This increment in Raman absorption considers a change in crystallinity of the material”.
  • By using glycerol, authors did not perform a calibration of the KD2-Pro, but a validation/checking of used experimental procedure. Have authors performed any kind of comparison with other standard fluids like water in the same temperature range? What did authors intended to say with “the thermal conductivity is verified up to 3 decimal points”? (please, better explain that sentence). Indicate also the literature reference they used to stablish such comparison.
  • The format/style of the graphs could be further improved (so that all of them looked similar).
  • A nomenclature table (or short list) summarizing the abbreviations used throughout the manuscript would be of great help for the reader.
  • In Figure 3, indicate what error bars stand for (standard deviations or so).
  • Please, clarify how enhancements in thermal conductivity have been calculated.
  • Authors mention in the abstract that toroidal carbon nanostructures could exhibit superb antiwear or friction reduction performance. However, authors do not perform such studies in this article. Do they plan to perform those analyses in the future?

Author Response

(The authors gave the same response as above.)

Reviewer 4 Report

The authors have prepare a draft about the thermal conductivity enhancements of a series of water-based and oil-based lubricants when carbon nanotories are added. Despite these results are promising, I recommend this manuscript to be rejected for publication in Nanomaterials. The performance of a lubricant is much more complicated than the measurment of the thermal conductivity. There are several considerations that authors should take into account:

  1. Regarding the thermal behavior of the mixtures: there are many other thermal properties that should have studied. TGA and DSC are mandatory.
  2. Characterization: FT-IR, viscosity, density,... where are all these measurements which are essential for a lubricant?
  3. Stability of the sample. Carbon nanostructures are likely to precipitate. It is mandatory to assess the stability of the mixtures and the possibility of using other additives to improve the dispersion of the nanophase.
  4. The tribological performance of the lubricants has not been attempted. These fluids may have some interest in tribology, but the authors should show it. Water-based lubricants are tricky since water is usually evaporated during the tribological tests due to the heat in the contact. I recommend the authors to perform immersion tests. Maybe the oil-based lubricants may be easier to study.

In addition, the self-citations of the authors is innapropiate and injustified. 

Therefore, I cannot support this manuscript to be published in Nanomaterials.

Author Response

(The authors gave the same response as above.)

Round 2

Reviewer 1 Report

Accept as is

Author Response

We thank the reviewer for his/her time and effort to review our manuscript. We appreciate his/her suggestion to Accept our manuscript as it is.

Regards

JT

Reviewer 3 Report

Authors have successfully performed the changes suggested by this reviewer. In my opinion, the article may be accepted in the present form.

Author Response

We thank the reviewer for his/her time and effort to review our manuscript. We appreciate his/her suggestion to Accept our manuscript in the present form.

Regards

JT

Reviewer 4 Report

Regrettably, I am not satisfied with the answers of the authors. They just have argued that due to COVID restrictions their paper cannot be improved. My opinion as expert is that the quality standards of the journal are not achieved. Their results are promising, but not complete. If they are not willing to make significant improvements in the manuscript, I suggest that the authors should send it to another journal with lower impact factor after decreasing the injustified number of self-citations.

Author Response

We appreciate Reviewer 4 for his/her time and effort to read and review our manuscript.

UV-Vis has been analyzed. UV-Vis spectra was performed for sample right after it was prepared (as-prepared, 0 h), and for sample after 10 weeks of sitting on the shelf.  We have included that in the manuscript together with Figure 3.

We understand there are broad opportunities for lubricants to explore, and that is something we are planning to do in the near future, but for now, we have included all the recommendations and comments in the manuscript.

Regards,